# Gene signatures associated with barrier dysfunction and infection in oral lichen planus identified by analysis of transcriptomic data

Phuc Thi-Duy Vo[1], Sun Shim Choi[2], Hae Ryoun Park[3], Ahreum Lee[1], Sung-Hee Jeong[4]*, Youngnim Choi[1]*

1 Department of Immunology and Molecular Microbiology, School of Dentistry and Dental Research Institute, Seoul National University, Seoul, Republic of Korea, 2 Division of Biomedical Convergence, College of Biomedical Science, Institute of Bioscience & Biotechnology, Kangwon National University, Chuncheon, Gangwon, Republic of Korea, 3 Department of Oral Pathology, Pusan National University School of Dentistry, Yangsan, Gyeongnam, Republic of Korea, 4 Department of Oral Medicine, Dental and Life Science Institute, Dental Research Institute, Pusan National University School of Dentistry, Yangsan, Gyeongnam, Republic of Korea

* drcookie@pusan.ac.kr (SHJ); youngnim@snu.ac.kr (YC)

**Data Availability Statement:** All data associated with this study are presented in the paper.

**Funding:** This study was supported by the National Research Foundation of Korea (Daejun, Korea)

## Abstract

Oral lichen planus (OLP) is one of the most prevalent oral mucosal diseases, but there is no cure for OLP yet. The aim of this study was to gain insights into the role of barrier dysfunction and infection in OLP pathogenesis through analysis of transcriptome datasets available in public databases. Two transcriptome datasets were downloaded from the Gene Expression Omnibus database and analyzed as whole and as partial sets after removing outliers. Differentially expressed genes (DEGs) upregulated in the dataset of OLP versus healthy epithelium were significantly enriched in epidermal development, keratinocyte differentiation, keratinization, responses to bacterial infection, and innate immune response. In contrast, the upregulated DEGs in the dataset of the mucosa predominantly reflected chemotaxis of immune cells and inflammatory/immune responses. Forty-three DEGs overlapping in the two datasets were identified after removing outliers from each dataset. The overlapping DEGs included genes associated with hyperkeratosis (upregulated *LCE3E* and *TMEM45A*), wound healing (upregulated *KRT17*, *IL36G*, *TNC*, and *TGFBI*), barrier defects (downregulated *FRAS1* and *BCL11A*), and response to infection (upregulated *IL36G*, *ADAP2*, *DFNA5*, *RFTN1*, *LITAF*, and *TMEM173*). Immunohistochemical examination of IL-36γ, a protein encoded by one of the DEGs *IL36G*, in control (n = 7) and OLP (n = 25) tissues confirmed the increased expression of IL-36γ in OLP. Collectively, we identified gene signatures associated with hyperkeratosis, wound healing, barrier defects, and response to infection in OLP. IL-36γ, a cytokine involved in both wound repair and antimicrobial defense, may be a possible therapeutic target in OLP.

through the grants 2018R1A5A2024418 and 2020R1A2C2007038 awarded to Youngnim Choi and 2019R1A2C1002350 awarded to Sun Shim Choi. The funders had no role in study design, data collection and analysis, decision to publish, or preparation of the manuscript.

**Competing interests:** The authors have declared that no competing interests exist.

## Introduction

Oral lichen planus (OLP), a variant of lichen planus, is a chronic T cell-mediated inflammatory disease of unknown etiology [1]. The global estimated prevalence of OLP in the general population is 1.01%, ranging from 0.47% to 1.74% with geographical differences. OLP occurs more frequently in over 40 years old, with a female predominance ratio of 1.5:1 [2]. Furthermore, OLP is defined by the World Health Organization as an oral potentially malignant disorder, with 2.28% malignant transformation [3]. OLP lesions are clinically classified into six types, reticular, papular, plaque, atrophic, erosive, and bullous, commonly affecting the buccal mucosae, tongue and gingival sites [1]. The histologic hallmarks of OLP include band-like lymphocytic infiltration, the presence of liquefaction degeneration in the basal cell layer, and hyperkeratosis with acanthosis [1]. In particular, the liquefaction degeneration reflects senescence of attacked basal cells and resembles the typical epithelial-mesenchymal transition alteration, thus, it might be related to malignant transformation [4–6].

Although several potential triggers, including genetic and psychological factors, systemic medications, trauma, and infections, have been suggested, the precise etiopathogenesis of OLP remains obscure [1]. Our group previously proposed a vicious cycle of epithelial barrier dysfunction and intracellular infection of epithelial basal cells with microbes as a potential model for OLP pathogenesis [7]. The increased expression of TLR1, TLR2, TLR3, TLR4, TLR7, TLR8, and TLR9 in OLP lesions [8] may indicate infection with microbes. Altered expression of several factors involved in epithelial differentiation and barrier function in OLP has also been reported [9]. Among the various inflammation-related cytokines detected in OLP lesions, tumor necrosis factor-α (TNFα), interferon-γ (IFN-γ), and interleukin 1-β (IL-1β) cause disruption of the epithelial tight junction barrier, but interleukin-17 (IL-17) maintains barrier integrity during epithelial injury through regulation of the tight junction protein occludin [10, 11].

In contrast to the studies that examine only a few molecules, transcriptome profiling provides a global snapshot for the molecular basis of a disease. To date, five groups have reported the various numbers of differentially expressed genes (DEGs) associated with OLP through transcriptomic analysis [12–16]. However, each group had slightly different aims and reported only some of the DEGs based on their own interest. To gain insights into the role of barrier dysfunction and infection in OLP pathogenesis, we performed analysis of two transcriptome datasets available in public databases and identified DEGs associated with aberrant keratinocyte differentiation and infection.

## Materials and methods

### Expression of transcriptomic data

Among the five previous studies, two transcriptome datasets, GSE52130 [13] and GSE38616 [14], deposited in public databases were included in the present study and downloaded from the National Center of Biotechnology Information Gene Expression Omnibus (GEO) database, a public repository for data storage (www.ncbi.nlm.nih.gov/geo). The GSE52130 dataset contained 7 OLP epithelial samples and 7 healthy epithelial samples based on the GPL10558 platform (Illumina HumanHT-12 V4.0 expression BeadChip), while the GSE38616 dataset was based on GPL6244 platform (Affymetrix Human Gene 1.0 ST Array) and consisted of 7 OLP mucosal samples and 7 healthy mucosal samples.

### DEGs analysis

R software (version 3.5.1) (http://www.r-project.org/) with the Bioconductor package (version 3.8) was used to perform background correction, quantile normalization, and probe summarization of

the raw data [17]. Student's t-tests were used to identify DEGs between OLP and healthy control samples. A $p$-value $< 0.05$ and |fold-change| $\geq 2$ were selected as the cutoff criteria for DEG screening. The Benjamini-Hochberg procedure was used to compute the false discovery rate (FDR)-corrected $p$-values, and $q$-values were reported. A $q$-value $< 0.05$ was considered statistically significant. Heat maps of DEGs combined with hierarchical clustering were generated with the hclust stats package in R (https://stat.ethz.ch/R-manual/R-patched/library/stats/html/hclust.html), and principal coordinate analysis (PcoA) plots were generated by using the factoMineR (http://factominer.free.fr) and rgl (https://r-forge.r-project.org/projects/rgl/) packages.

### Gene Ontology (GO) enrichment analysis

The online software Database for Annotation, Visualization, and Integrated Discovery (DAVID; version 6.8; http://david.abcc.ncifcrf.gov) was used to analyze functional biological processes for all datasets of DEGs based on the GO database (http://www.geneontology.org/). A $p$-value $< 0.05$ and a number of involved genes $\geq 2$ were selected as the cutoff criteria for GO biological term screening.

### Tissue samples and immunohistochemistry

This study was performed following the principles of the Declaration of Helsinki and was approved by the Pusan National University Dental Hospital (Busan, Korea) Institutional Review Board (IRB) (No. PNUDH-2019-024). Sections of formalin-fixed paraffin-embedded biopsy samples of 25 OLP patients and 7 patients diagnosed with other oral diseases were obtained from a tissue bank at the Pusan National University Dental Hospital.

For immunohistochemical staining, sections were deparaffinized and rehydrated followed by antigen retrieval by boiling in sodium citrate buffer for 10 min. Sections were then incubated with anti-IL-1F9 (dilution 1:50,000; Invitrogen, Carlsbad, CA, USA) or anti-IL-36Ra (dilution 1:30; Proteintech Group, Inc. Rosemont, IL, USA) antibodies at 4°C overnight, followed by incubation with horseradish peroxidase-conjugated secondary antibodies (dilution 1:250; Santa Cruz Biotechnology, Santa Cruz, CA, USA) at room temperature for 1 h. The bound antibody signals were visualized using an Envision System (DAKO, Hamburg, Germany) with 3,3'-diaminobenzidine as chromogen to yield brown-colored signals on the tissue sections. In each sample, four areas per epithelium and lamina propria were photographed at 200x magnification using an Automated Upright Microscope System (Leica Biosystem, Germany). After coding the images, IHC signals were blindly quantified using ImageJ software (National Institute of Mental Health, Bethesda, MD, USA).

### Statistical analysis

Student's t-tests were used to identify DEGs between OLP and healthy control samples. The t-tests and Benjamini-Hochberg procedure were performed using the R software. The Mann-Whitney U-test and Spearman's rank correlation test were used to analyze the immunohistochemistry data, and receiver operating characteristic (ROC) analysis was performed using SPSS Statistics 26 software (SPSS Inc., Chicago, IL, USA). The significance level was set at $p$ or $q < 0.05$.

## Results

### Removing outliers increased the number of DEGs in each dataset, and 43 overlapping DEGs were identified

Gene expression profiles can provide new insights into the molecular pathophysiology of OLP. Since transcriptome analysis is usually performed using a small sample size, we asked if there

are common DEGs in two independent studies using the GSE52130 and GSE38616 datasets available in the NCBI GEO database.

In the GSE52130 dataset that analyzed the transcriptomes of epithelium obtained from seven OLP patients and seven healthy subjects, a total of 14,692 transcripts were present. Among these, 200 DEGs (137 upregulated and 63 downregulated) were identified in the comparison of OLP versus healthy samples using the criteria $p < 0.05$, |fold-change| $\geq 2$, and $q < 0.05$ (S1 Table). Removing outliers is a common method to strengthen the power of detecting DEGs [18]. Cluster analysis of the 14 transcriptomes revealed that three OLP and one healthy sample did not cluster with the other samples in each corresponding group (Fig 1A and 1B). After removing these four outliers, the OLP and healthy samples clustered into two distinct groups in a principal component analysis (PcoA) plot (Fig 1C). From the analysis of these partial sets, 444 DEGs (257 upregulated and 187 downregulated) were obtained (S2 Table).

The GSE38616 dataset included a total of 22,195 transcripts obtained from seven OLP and seven healthy mucosae. Among those, 33 DEGs (22 upregulated and 11 downregulated) were found in the comparison of OLP versus healthy samples using the criteria $p < 0.05$ and |fold-change| $\geq 2$, and none of the genes passed a Benjamini-Hochberg FDR correction test (S3 Table). Cluster analysis of the 14 transcriptomes indicated that two OLP and three healthy samples did not cluster with the other samples in each corresponding group (Fig 1D and 1E). From the mucosal partial set that excluded the five outliers (Fig 1F), 348 DEGs (294 upregulated and 54 downregulated) were obtained using the criteria $p < 0.05$, |fold-change| $\geq 2$ and $q < 0.05$.

To identify common DEGs of the two datasets, the DEG lists were compared. When the DEGs out of the epithelial whole dataset were compared with those of the mucosal whole dataset, only 1 common DEG was found (Fig 1G): *KLK12*, a gene encoding a secreted serine protease involved in angiogenesis, was upregulated by 4.2-fold in the epithelium ($p = 0.001$, $q = 0.018$) and 2.6-fold in the mucosa of OLP subjects ($p = 0.028$, $q = 0.43$). In the comparison of the DEGs of the two partial datasets, 43 overlapping DEGs (23 upregulated and 20 downregulated in both sets) were identified (Fig 1H and Table 1). There was no common DEG that was upregulated in one set but downregulated in the other set. The top overlapping upregulated DEGs (fold change > 10) included *LCE3E*, *KRT17*, *TMEM45A*, and *IL-36G*, which encode late cornified envelope protein 3E, keratin 17, transmembrane protein 45A, and interleukin-36 gamma (IL-36γ, also known as IL-1F9), respectively.

## GO analyses revealed diverse biological processes enriched in OLP lesions

To explore the functional biological processes in OLP, enrichment analyses were performed using the DAVID. The upregulated DEGs of the whole and partial epithelial sets were significantly enriched in epidermal development, keratinocyte differentiation, keratinization, responses to bacterial infection, and innate immune response, while downregulated DEGs mainly involved oxidation-reduction process and responses to several ions (Fig 2A and 2B, S5 and S6 Tables). On the other hand, the enriched processes of upregulated DEGs in the whole and partial mucosal sets predominantly reflected chemotaxis of immune cells and inflammatory, innate, and adaptive responses by the infiltrated cells. The enriched processes in the partial mucosal set also included responses to LPS and antigen presentation via MHC class II (Fig 2C and 2D, S7 and S8 Tables). In the 43 overlapping DEGs, the upregulated genes were associated with cell adhesion and proliferation, while the downregulated genes were linked to the oxidation-reduction process and receptor tyrosine phosphatase signaling pathway (Fig 2E, S9 Table).

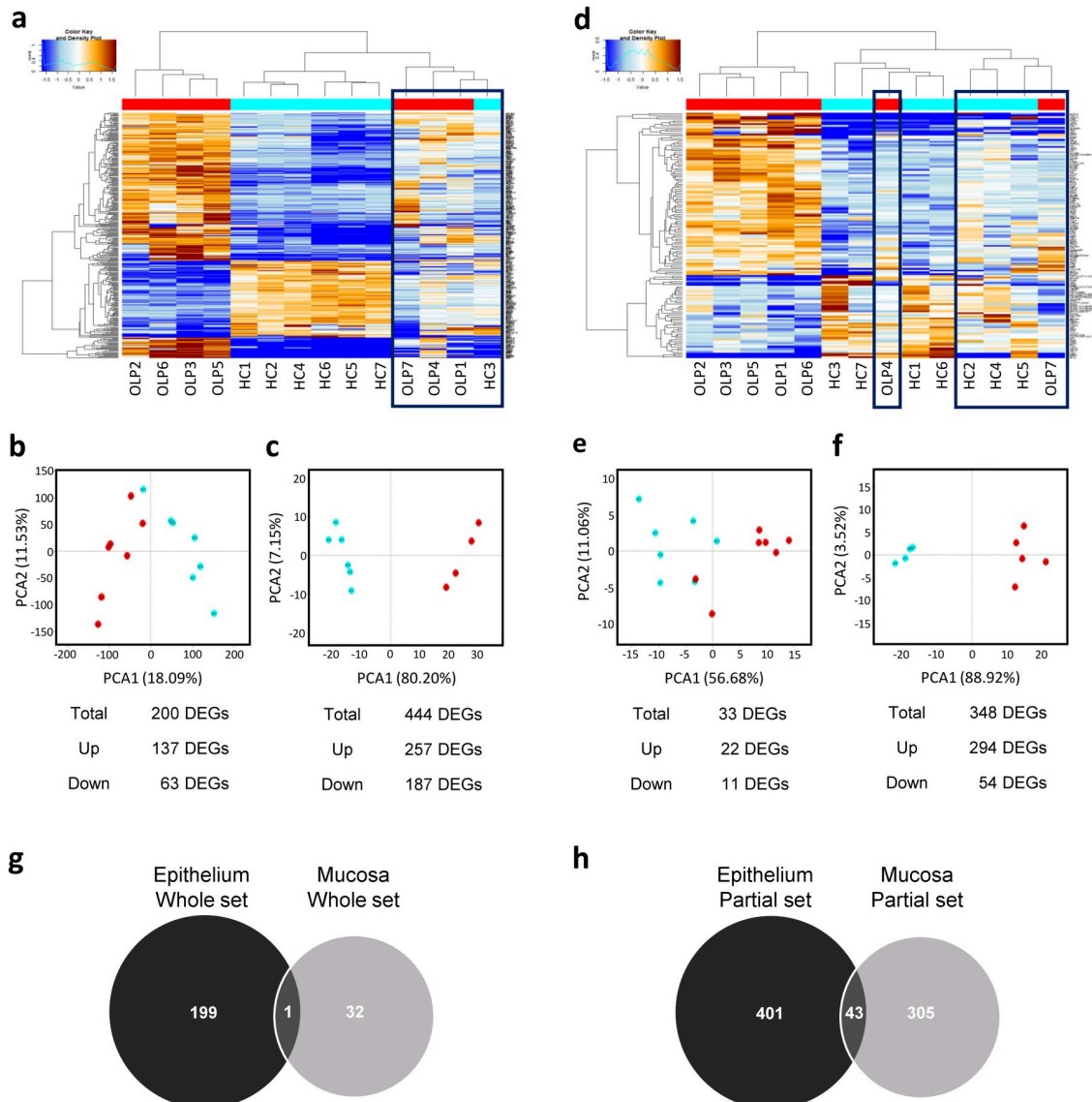

**Fig 1. Differentially expressed gene (DEG) analysis using the two datasets GSE52130 and GSE38616.** (a-c) GSE52130, the transcriptome of the epithelium, and (d-f) GSE38616, the transcriptome of the mucosa, were downloaded from the GEO database and analyzed. (a, d) Heat maps of DEGs combined with hierarchical clustering. The color change from brown to blue represents the change from upregulation to downregulation. Black squares marks outliers removed in the partial sets. (b,c,e,f) Principal coordinate analysis plots of the whole (b,e) and partial (c,f) datasets. (g,h) Venn diagram illustrating the number of DEGs in the two whole (g) and partial (h) datasets. The black circle represents the GSE52130 dataset, and the gray circle represents the GSE38616 dataset. The intersection of the 2 circles indicates the overlapping DEGs between the two datasets.

## Expression of IL-36γ but not that of IL-36 receptor antagonist (Ra) is increased in OLP tissues

To validate the results of our bioinformatic analysis at the protein level, *IL-36G* was chosen among the top five upregulated overlapping DEGs because a secretory cytokine is an attractive therapeutic target compared with intracellular proteins. The activities of IL-36 cytokines, including IL36α, IL-36β, and IL-36γ, are highest at barrier sites (skin, lung, and intestine) and believed to play an important role in maintaining epithelial homeostasis [19]. The expression

**Table 1. Overlapping DEGs between two partial data sets of epithelium and mucosa.**

| Gene Symbol | Epithelium | | | Mucosa | | |
|---|---|---|---|---|---|---|
| | Fold-change | *p*-value | *q*-value | Fold-change | *p*-value | *q*-value |
| LCE3E | 37.6 | 1.7E-07 | 0.000 | 25.5 | 1.5E-04 | 0.027 |
| KRT17 | 36.7 | 2.1E-04 | 0.005 | 10.6 | 6.7E-04 | 0.039 |
| TMEM45A | 18.7 | 1.1E-04 | 0.007 | 11.4 | 8.1E-05 | 0.024 |
| IL36G | 14.9 | 1.1E-06 | 0.001 | 17.0 | 5.1E-05 | 0.020 |
| ADAP2 | 6.2 | 2.1E-08 | 0.000 | 3.8 | 1.3E-04 | 0.025 |
| ERP27 | 4.7 | 5.7E-06 | 0.001 | 3.1 | 3.4E-05 | 0.018 |
| RFTN1 | 4.5 | 3.7E-04 | 0.007 | 2.6 | 1.2E-04 | 0.025 |
| FEZ1 | 4.2 | 4.2E-03 | 0.026 | 2.2 | 1.2E-03 | 0.046 |
| CCND2 | 3.8 | 9.1E-08 | 0.000 | 3.3 | 4.2E-04 | 0.034 |
| TNC | 3.6 | 1.2E-04 | 0.004 | 7.2 | 1.1E-03 | 0.046 |
| TGFBI | 3.1 | 8.4E-03 | 0.039 | 3.6 | 2.5E-04 | 0.031 |
| DFNA5 | 3.0 | 2.7E-03 | 0.020 | 2.2 | 7.7E-04 | 0.041 |
| LPXN | 2.7 | 1.8E-03 | 0.016 | 3.9 | 1.4E-05 | 0.018 |
| NABP1 | 2.5 | 6.4E-03 | 0.033 | 2.4 | 1.1E-03 | 0.045 |
| LITAF | 2.4 | 7.9E-04 | 0.010 | 2.3 | 2.7E-04 | 0.032 |
| FAM167A | 2.3 | 4.7E-04 | 0.008 | 2.2 | 8.8E-04 | 0.042 |
| SLC39A6 | 2.3 | 3.6E-03 | 0.006 | 2.6 | 2.9E-05 | 0.018 |
| GPR137B | 2.2 | 2.0E-03 | 0.017 | 2.7 | 1.7E-04 | 0.027 |
| ANTXR2 | 2.1 | 1.2E-02 | 0.049 | 3.2 | 1.2E-03 | 0.047 |
| FAM69A | 2.1 | 3.9E-05 | 0.003 | 2.5 | 5.4E-04 | 0.036 |
| TMEM173 | 2.1 | 3.8E-03 | 0.024 | 2.3 | 1.4E-03 | 0.049 |
| INPP4B | 2.0 | 4.7E-03 | 0.028 | 2.1 | 3.1E-04 | 0.033 |
| UBASH3B | 2.0 | 6.0E-03 | 0.032 | 2.0 | 1.1E-04 | 0.025 |
| PTPRF | -2.0 | 3.9E-03 | 0.024 | -2.2 | 2.7E-04 | 0.032 |
| RAPGEFL1 | -2.1 | 8.3E-05 | 0.004 | -4.4 | 1.3E-03 | 0.048 |
| CBR1 | -2.2 | 9.9E-04 | 0.012 | -2.7 | 2.6E-04 | 0.031 |
| PLLP | -2.3 | 1.2E-04 | 0.002 | -2.8 | 5.3E-06 | 0.013 |
| FRAS1 | -2.4 | 8.2E-07 | 0.000 | -2.6 | 3.5E-04 | 0.034 |
| AIM1L | -2.5 | 4.4E-04 | 0.019 | -2.2 | 3.3E-04 | 0.034 |
| MGST2 | -2.5 | 5.9E-04 | 0.009 | -2.6 | 4.0E-04 | 0.034 |
| PTN | -2.5 | 1.1E-09 | 0.000 | -3.9 | 5.5E-05 | 0.021 |
| CYP11A1 | -2.8 | 1.4E-06 | 0.001 | -2.7 | 5.4E-04 | 0.036 |
| SCIN | -3.1 | 7.5E-05 | 0.004 | -8.6 | 6.5E-05 | 0.022 |
| HMGCS1 | -3.3 | 1.8E-05 | 0.002 | -4.2 | 2.1E-04 | 0.029 |
| ZBTB7C | -3.5 | 3.7E-06 | 0.001 | -3.3 | 1.4E-03 | 0.049 |
| CYP4F12 | -3.6 | 1.9E-03 | 0.016 | -3.2 | 6.9E-04 | 0.039 |
| BCL11A | -3.7 | 4.0E-05 | 0.003 | -2.4 | 4.6E-04 | 0.035 |
| FGFR3 | -3.7 | 1.0E-04 | 0.004 | -3.0 | 1.4E-03 | 0.049 |
| MAOA | -3.9 | 2.5E-04 | 0.006 | -3.1 | 1.1E-05 | 0.017 |
| PGD | -4.1 | 1.6E-06 | 0.001 | -2.9 | 6.9E-04 | 0.039 |
| WNK4 | -4.4 | 8.0E-04 | 0.010 | -2.2 | 8.4E-04 | 0.042 |
| ALDH3A1 | -4.8 | 1.0E-02 | 0.044 | -5.5 | 1.1E-03 | 0.046 |
| ETNK2 | -5.1 | 4.8E-05 | 0.003 | -12.1 | 1.5E-04 | 0.027 |

of IL-36γ was examined by immunohistochemistry using tissue sections of 25 OLP cases and 7 control cases with other oral diseases that were chosen based on the histopathology and availability. Since the function of IL-36γ is antagonized by IL-36Ra encoded by *IL-36RN*, the

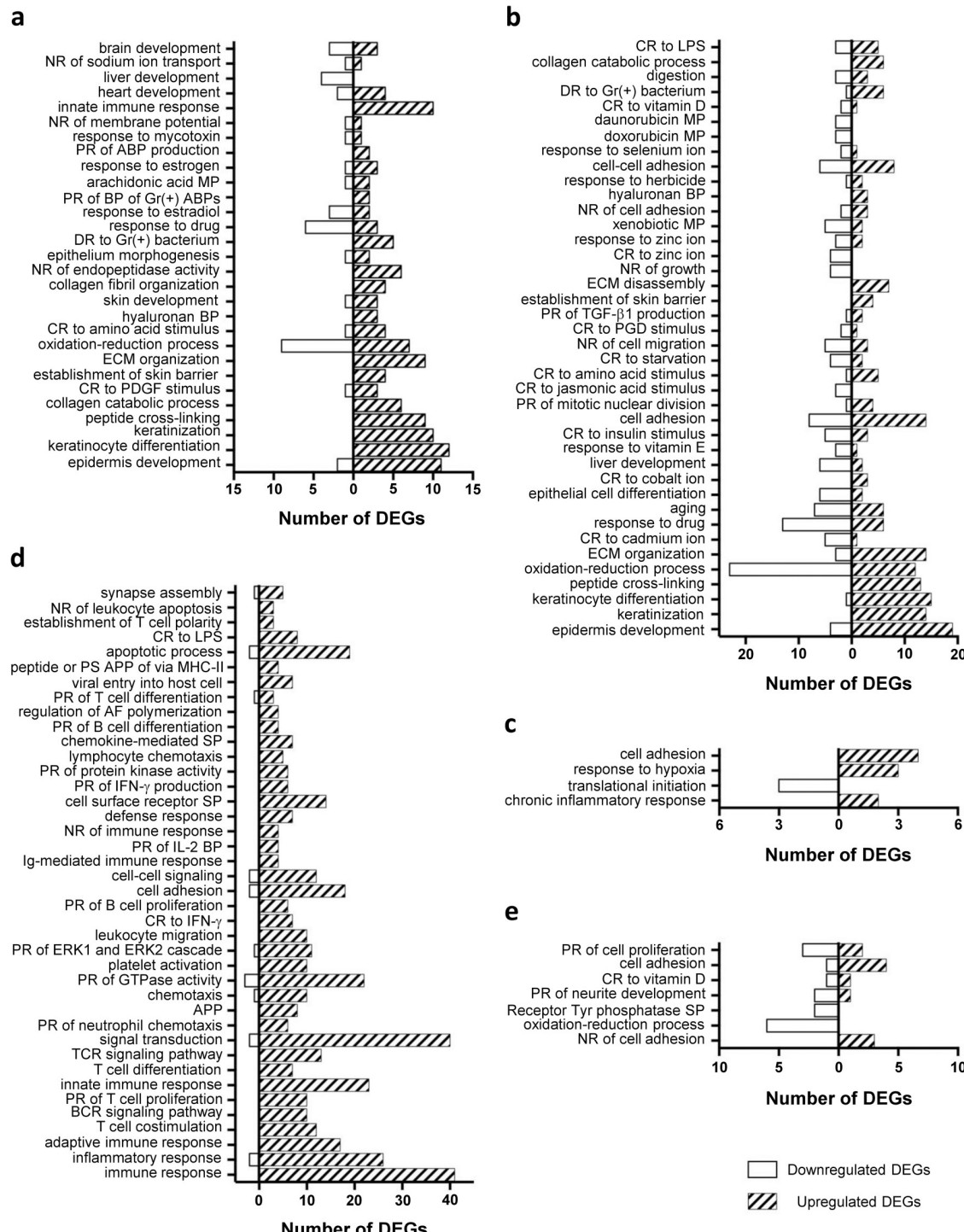

**Fig 2. GO biological process terms enriched in DEGs.** (a) GO terms enriched in the epithelium whole dataset. (b) Top 40 GO terms enriched in the epithelium partial dataset. (c) GO terms enriched in the mucosa whole dataset. (d) Top 40 GO terms enriched in the mucosa partial dataset. (e) GO terms enriched in the overlapping DEGs between the two partial datasets. The GO terms are ordered with the smallest *p*-value from the bottom of each graph. ABP: antibacterial peptide, AF: actin filament, APP: antigen processing and presentation, BP: biosynthetic process, CR: cellular response, DR: defense response, ECM: extracellular matrix, LPS: lipopolysaccharide, MP: metabolic process, NR: negative regulation, PDGF: platelet-derived growth factor, PGD: prostaglandin D, PR: positive regulation, PS: polysaccharide, SP: signaling pathway.

**Table 2. Clinicopathological characteristics of OLP and control patients.**

| Group | No. | Sex | Age | Duration (year) | REU score | Histopathological diagnosis | Site of lesions |
|---|---|---|---|---|---|---|---|
| **OLP** | 1 | F | 45 | 0.3 | 4 | OLP | Buccal mucosa |
| | 2 | F | 55 | 0.3 | 10 | Lichenoid inflammation | Buccal mucosa |
| | 3 | M | 62 | 1 | 2.5 | OLP | Vestibular mucosa |
| | 4 | M | 65 | 0.4 | 2.5 | OLP | Buccal mucosa |
| | 5 | F | 65 | 2 | 4.5 | OLP | Edentulous ridge |
| | 6 | M | 47 | 0.8 | 2.5 | OLP | Gingiva |
| | 7 | F | 55 | 1 | 5.5 | OLP | Buccal mucosa |
| | 8 | F | 44 | 4 | 1 | Chronic inflammation, c/w OLP. | Buccal mucosa |
| | 9 | F | 71 | 1 | 7 | OLP | Retromolar area |
| | 10 | F | 64 | 3 | 6.5 | Chronic inflammation, c/w OLP | Buccal mucosa |
| | 11 | M | 63 | 6 | 4 | Chronic inflammation, c/w OLP | Buccal mucosa |
| | 12 | F | 70 | 1.3 | 6 | OLP | Buccal mucosa |
| | 13 | M | 57 | 1 | 6.5 | OLP | Buccal mucosa |
| | 14 | F | 55 | 0.3 | 4 | OLP | Buccal mucosa |
| | 15 | F | 54 | 1 | 2.5 | Chronic inflammation, c/w OLP | Gingiva |
| | 16 | F | 64 | 20 | 4 | OLP | Buccal mucosa |
| | 17 | F | 54 | 10 | 4 | OLP | Buccal mucosa |
| | 18 | M | 57 | 1.5 | – | Lichenoid inflammatory infiltration, epithelial separation | Buccal mucosa |
| | 19 | F | 60 | 1 | 3 | Chronic inflammation, c/w OLP | Gingiva |
| | 20 | F | 53 | 4 | 4.5 | Polymorphic lymphocyte infiltration, c/w OLP | Buccal mucosa |
| | 21 | F | 69 | 0.5 | 1 | OLP | Buccal mucosa |
| | 22 | F | 43 | 0.2 | 1 | OLP | Buccal mucosa |
| | 23 | F | 55 | 1 | 2.5 | OLP | Tongue lateral border |
| | 24 | F | 57 | 0.8 | 6 | OLP | Buccal mucosa |
| | 25 | M | 70 | 1 | 4 | c/w OLP | Buccal mucosa |
| **Control** | 1 | M | 59 | 0.3 | – | Chronic inflammation with acanthosis | Buccal gingiva |
| | 2 | M | 41 | 1 | – | Acanthosis with hyperkeratosis and fibrosis | Retromolar area |
| | 3 | F | 8 | – | – | Chronic inflammation | Lower lingual frenum |
| | 4 | M | 68 | 0.08 | – | Fibroma | Tongue tip |
| | 5 | M | 54 | 1 | – | Mild inflammation with acanthosis and pigmentation | Tongue lateral border |
| | 6 | F | 77 | 0.08 | – | Epulis fissuratum | Maxillary vestibule |
| | 7 | F | 48 | 0.2 | – | Hyperkeratosis and parakeratosis | Tongue |

OLP: oral lichen planus, REU: reticulation/keratosis; erythema; ulceration, c/w: consistent with.

expression of IL-36Ra was also examined in parallel, although *IL-36RN* was not included in the overlapping DEGs.

The participant population showed a wide range of clinicopathological features, as described in Table 2. Among the OLP patients, females accounted for almost three fourths of the participants. The onset from the biopsy time ranged from 2 months to 20 years. Clinical severity at the biopsy site, which was evaluated with reticulation, erythema, and ulceration scores, ranged from 1 to 10. The most common site of the biopsy was the buccal mucosa, followed by gingivae. The histopathological diagnoses of the control tissues included chronic inflammation, acanthosis, fibroma, epulis fissuratum, and hyperkeratosis, but the histopathological abnormalities observed in control tissues were limited.

As depicted in Fig 3A, IL-36γ was expressed in epithelial cells (asterisks) throughout the epithelium of OLP tissue and also in the infiltrated immune cells (arrows). The signal

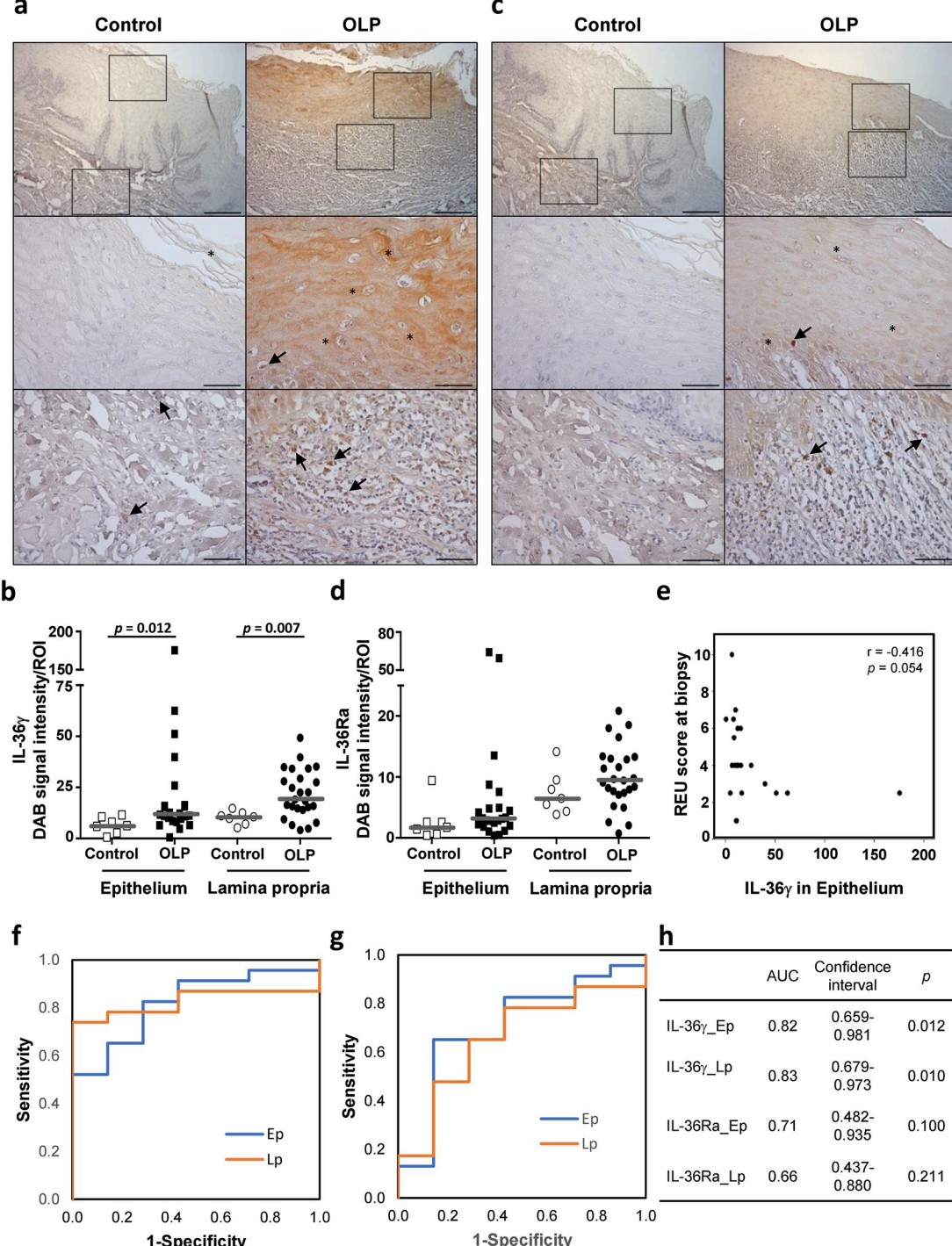

**Fig 3. Immunohistochemical detection of IL-36γ and IL-36Ra in OLP versus control tissue sections.** Sections of OLP (n = 25) and control (n = 7) tissues were subjected to immunohistochemical detection of IL-36γ (a) and IL-36Ra (c), and the signal intensities in the epithelium and mucosa were measured by ImageJ (b, d). Asterisks and arrows depict IL-36γ expression in epithelial cells and immune cells, respectively. The expression levels in the presented images are equivalent to the median value of each group (low magnification x100, scale bar = 200 μm; high magnification x400, scale bar = 50 μm). (e) Correlation plot between the levels of IL-36γ expression in the epithelium and clinical severity scores. (f, g) Receiver operating characteristic (ROC) curves of IL-36γ (f) and IL-36Ra (g) expressions in the epithelium (Ep) and lamina propria (Lp). (h) The area under curve (AUC) and significance of ROC curves shown in f and g.

intensities of IL-36γ were higher in the epithelium and the lamina propria of OLP samples than in those of control samples (Fig 3B, $p = 0.012$ and $p = 0.007$, respectively). Particularly, in the lamina propria, 76% of OLP tissues presented a stronger IL-36γ signal than the control tissue with the highest expression level. The expression pattern of IL-36Ra was similar to that of IL-36γ (Fig 3C). The median expression levels of IL-36Ra in OLP tissues were higher than those in control tissues in both the epithelium and lamina propria, but the differences were not significant ($p > 0.05$) (Fig 3D). There was no inter-group difference in IL-36γ/IL-36Ra ratio, either. Interestingly, the clinical severity scores tended to have a negative correlation with the IL-36γ expression levels in the epithelium ($r_s = -0.416$, $p = 0.054$) (Fig 3E) but not those in the lamina propria ($r_s = -0.053$, $p = 0.806$).

ROC curve analysis revealed that the expression levels of IL-36γ both in the epithelium and lamina propria could differentiate OLP from disease controls based on the area under curve (AUC > 0.7, $p < 0.05$). In contrast, IL-36Ra was not a significant marker (Fig 3F–3H).

## Discussion

OLP is one of the most prevalent oral mucosal diseases, but there is no cure for OLP yet. To gain insights into the role of barrier dysfunction and infection in OLP pathogenesis, two transcriptome datasets available in the public database were analyzed, and DEGs associated with aberrant keratinocyte differentiation and infection were identified.

In the current study, we analyzed the GEO data as whole and as partial sets after removing outliers. The variations in transcriptome profiles revealed by cluster analysis (Fig 1A and 1D) may be attributed to differences in the clinical types of OLP, the composition of infiltrated immune cells, or the quality of RNA. The subject-to-subject variations in the mucosal dataset were particularly substantial, yielding only 33 DEGs that did not pass the Benjamini-Hochberg FDR correction test. By removing five outliers in the dataset, we identified 348 DEGs that satisfied the Benjamini-Hochberg FDR correction test at $q < 0.05$. Similarly, DEGs in the OLP epithelium increased from 200 to 444 by removing four outliers. Furthermore, we identified 43 DEGs overlapping in the two partial sets of the epithelium and mucosa. These overlapping DEGs may reflect bonafide changes occurring in the epithelium of typical OLP cases.

GO analyses revealed that the most enriched biological processes involving the upregulated DEGs in the epithelium (both the whole and partial datasets) were epidermal development, keratinocyte differentiation, keratinization, and peptide crosslinking. These biological processes reflect the hyperkeratosis with acanthosis observed in OLP. Interestingly, the biological process of skin barrier establishment was also enriched with upregulation of *ALOX12B*, *ALOXE3*, *FLG*, and *KRT16* (S5 and S6 Tables). Deficiency or mutation in these genes results in perturbation of skin barrier function [20–22]. However, upregulated *FLG* reflects hyperkeratosis [23], while *ALOX12B*, *ALOXE3*, and *KRT16* are wound-activated genes in the oral mucosa, suggesting an ongoing wound repair process [24]. The GO terms wound healing and response to wounding were also enriched in the partial sets of the epithelium and mucosa, respectively (S6 and S8 Tables).

Among the 43 overlapping DEGs identified in the partial sets of the epithelium and mucosa, high *LCE3E* and *TMEM45A* expression is associated with epidermal keratinization [25], and upregulation of *IL36G*, *TNC*, *TGFBI*, and *KRT17* has been observed in wounded oral mucosa or skin [24, 26, 27]. In particular, *KRT17* is upregulated together with *KRT16* in response to a barrier breach, and their products keratin 16 and 17 contribute to hyperproliferation and innate immune activation of keratinocytes as barrier alarmin molecules [28]. Moreover, downregulation of *FRAS1* and *BCL11A* among the 43 overlapping DEGs is associated with barrier defects. Fraser syndrome protein 1 (FRAS1) encoded by *FRAS1* is one of the three Fraser

syndrome-associated proteins that form a mutually stabilized protein complex at the basement membrane and anchor the basement membrane to its underlying mesenchyme [29]. Deficiency or mutation in any individual gene leads to blister formation [30]. Downregulated *FRAS1* may reflect the detachment of the epithelium from lamina propria that is often observed in OLP. BCL11A is a transcription factor regulating lipid metabolism and terminal differentiation of keratinocytes, including profilaggrin processing, that are critical for the epidermal permeability barrier [31]. Without adequate function of BCL11A, the hyperkeratotic epithelium observed in OLP may present permeability barrier defects. Danielsson et al. interpreted the upregulated expression of keratinocyte late differentiation genes, including *LOR*, *CDSN*, *LCE*, and *FLG*, as representative of a strengthened epithelial barrier [14]. However, we propose that the gene signature identified in the two OLP datasets suggests chronic wounds and epithelial barrier dysfunction.

Among the enriched GO terms identified in the epithelial dataset and mucosa partial set, defense responses to both gram-positive and negative bacteria, positive regulation of antibacterial peptides active against gram-positive bacteria, positive regulation of antibacterial peptide production, cellular response to lipopolysaccharide (LPS), Toll-like receptor 3 signaling pathway, and antigen processing and presentation of exogenous peptide antigen via MHC class II (Fig 2, S6 and S8 Tables) indicated potential microbial infection in OLP. In addition, several DEGs, including *IL36G*, *ADAP2*, *DFNA5*, *RFTN1*, *LITAF*, and *TMEM173*, that were commonly upregulated in the partial sets of the epithelium and mucosa are associated with the response to infection. For example, ADAP2 mediates the antiviral effects of type I IFN against RNA viruses [32]; gasdermin E encoded by *DFNA5* is cleaved by caspase-3 and induces pyroptosis, an effective defense mechanism against intracellular bacteria [33]; and *TMEM173* encodes stimulator of interferon genes (STING), which serves as a critical signaling adaptor in the innate immune response to cytosolic DNA and RNA derived from pathogens [34]. We recently reported the detection and isolation of *Escherichia coli* from OLP tissues [35]. Therefore, the defense response to gram-negative bacterium and the response to LPS are particularly interesting. LITAF, an LPS-induced TNF transcription factor, is induced by LPS from *E. coli* to mediate inflammatory cytokine expression [36]. RFTN1, also known as Raftlin, mediates the LPS-induced endocytosis of TLR4 required for IFN-β production [37]. Moreover, *IL36G* is one of the canonical molecules triggered by infection with uropathogenic *E. coli* in the mouse bladder [38].

IL-36 cytokines are produced predominantly by keratinocytes, but also by immune cells, such as dendritic cells, macrophages, T cells, and plasma cells under inflammation [19]. Expression of both IL-36γ and IL-36Ra by keratinocytes and infiltrated immune cells was observed in OLP lesions (Fig 3A and 3C). The expression of IL-36 cytokines in keratinocytes is upregulated by many cytokines, including TNFα, IL-17, IL-22, IFNγ, and IL-36 itself, and by TLR agonists [39]. Interestingly, except *IL36G* and *IL1* (S1–S4 Tables), no other inflammatory cytokines were identified as DEGs in the datasets analyzed in this study. Therefore, the increased expression of *IL36G* observed in OLP lesions could be caused by TLR agonists. Higher expression of IL-36γ in OLP lesions than control tissues was confirmed by immunohistochemistry, despite the presence of various histological abnormalities in the control tissues due to other oral diseases (Fig 3B). However, the difference in the levels of IL-36Ra was not significant (Fig 3D). Furthermore, ROC analysis revealed that IL-36 γ can serve as a biomarker to differentially diagnose OLP from other oral mucosal diseases (Fig 3H).

IL-36γ is also known as a biomarker for psoriasis. In contrast to the situation in OLP, however, the overexpression of IL-36γ is limited to the epithelium, and IL-36γ expression positively correlates with disease severity in psoriasis [40]. Buhl and Wenzel suggested that a positive feedback loop between IL-36 cytokines and IL-17 contributes to epidermal thickening

observed in psoriasis [19]. Unexpectedly, the level of IL-36γ in the epithelium presented a tendency toward a negative correlation with OLP severity (Fig 3G). IL-36γ expression is induced in keratinocytes by bacterial, fungal, or herpes simplex virus infection and has a leading role in the clearance of infected microbes by inducing antimicrobial peptides, inflammatory cytokines, and chemokines [41]. It has been shown in mice that skin injury-induced IL-36γ promotes wound healing via REG3A [42]. Likewise, as IL-36γ is substantially upregulated in the human oral mucosa during wound healing [24], upregulated IL-36γ could be beneficial for wound healing and infection control in OLP lesions. The function of IL-36γ has been extensively studied in the skin, lung, and intestine but not in the oral mucosa. The precise role of IL-36γ in the pathophysiology of OLP needs further clarification.

In conclusion, we identified gene signatures associated with hyperkeratosis, wound healing, barrier defects, and response to infection in OLP. Whether infection is the result of barrier defects/wounds or the cause of chronic wounds is not clear, but breaking this vicious cycle seems to be important. IL-36γ, a cytokine involved in both wound repair and antimicrobial defense, may be a possible therapeutic target in OLP.

## Supporting information

**S1 Table. Differentially expressed genes (DEGs) in the epithelium whole dataset.**
(PDF)

**S2 Table. Differentially expressed genes (DEGs) in the epithelium partial dataset.**
(PDF)

**S3 Table. Differentially expressed genes (DEGs) in the mucosa whole dataset.**
(PDF)

**S4 Table. Differentially expressed genes (DEGs) in the mucosa partial dataset.**
(PDF)

**S5 Table. Gene Ontology biological process terms enriched in the epithelium whole dataset.**
(PDF)

**S6 Table. Gene Ontology biological process terms enriched in the epithelium partial dataset.**
(PDF)

**S7 Table. Gene Ontology biological process terms enriched in the mucosa whole dataset.**
(PDF)

**S8 Table. Gene Ontology biological process terms enriched in the mucosa partial dataset.**
(PDF)

**S9 Table. Gene Ontology biological process terms enriched in the common DEGs of the epithelium and mucosa partial datasets.**
(PDF)

## Author Contributions

**Conceptualization:** Sung-Hee Jeong, Youngnim Choi.

**Data curation:** Phuc Thi-Duy Vo, Hae Ryoun Park, Sung-Hee Jeong.

**Formal analysis:** Phuc Thi-Duy Vo.

**Funding acquisition:** Sun Shim Choi, Youngnim Choi.

**Investigation:** Phuc Thi-Duy Vo.

**Methodology:** Sun Shim Choi, Ahreum Lee.

**Supervision:** Sun Shim Choi, Youngnim Choi.

**Writing – original draft:** Phuc Thi-Duy Vo.

**Writing – review & editing:** Sun Shim Choi, Hae Ryoun Park, Ahreum Lee, Sung-Hee Jeong, Youngnim Choi.

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
