## [Decision Letter · Decision Letter 0]

12 Aug 2021

PONE-D-21-21443

Gene signatures associated with barrier dysfunction and infection in oral lichen planus identified by meta-analysis of transcriptomic data

PLOS ONE

Dear Dr. Choi,

Thank you for submitting your manuscript to PLOS ONE. After careful consideration, we feel that it has merit but does not fully meet PLOS ONE’s publication criteria as it currently stands. Therefore, we invite you to submit a revised version of the manuscript that addresses the points raised during the review process.

We look forward to receiving your revised manuscript.

Kind regards,

Kanhaiya Singh, Ph.D

Academic Editor

PLOS ONE

Journal Requirements:

“This study was supported by the National Research Foundation of Korea (Daejun, Korea) through the grants 2018R1A5A2024418 and 2020R1A2C2007038 awarded to Youngnim Choi and 2019R1A2C1002350 awarded to Sun Shim Choi.”

“This study was supported by the National Research Foundation of Korea (Daejun, Korea) through the grants 2018R1A5A2024418 and 2020R1A2C2007038 awarded to Youngnim Choi and 2019R1A2C1002350 awarded to Sun Shim Choi.”

We note that you have provided funding information within the Acknowledgements. Please note that funding information should not appear in the Acknowledgments section or other areas of your manuscript. We will only publish funding information present in the Funding Statement section of the online submission form.

“This study was supported by the National Research Foundation of Korea (Daejun, Korea) through the grants 2018R1A5A2024418 and 2020R1A2C2007038 awarded to Youngnim Choi and 2019R1A2C1002350 awarded to Sun Shim Choi.”

Additional Editor Comments:

Although the reviewers found the study interesting, they have recommended to revise this manuscript in order to have more clarity in results. Also the reason to select IL-36G out of several overlapping DEGs identified needs to be addressed.

Reviewers' comments:

Reviewer's Responses to Questions

**Comments to the Author**

1. Is the manuscript technically sound, and do the data support the conclusions?

Reviewer #1: Partly

Reviewer #2: Yes

Reviewer #3: Yes

Reviewer #4: Yes

2. Has the statistical analysis been performed appropriately and rigorously? 

Reviewer #1: Yes

Reviewer #2: Yes

Reviewer #3: Yes

Reviewer #4: Yes

3. Have the authors made all data underlying the findings in their manuscript fully available?

Reviewer #1: Yes

Reviewer #2: No

Reviewer #3: Yes

Reviewer #4: Yes

4. Is the manuscript presented in an intelligible fashion and written in standard English?

Reviewer #1: Yes

Reviewer #2: Yes

Reviewer #3: Yes

Reviewer #4: Yes

5. Review Comments to the Author

Reviewer #1: While going through the manuscript, I came through some lacking which have been addresses below:

1. I request the author to give logical and sound reasons for selection of IL-36G out of several overlapping DEGs identified. Out of two cytokines IL-36G and IL-1, author has chosen only IL-36G, I request to provide reason for selection of IL-36G.

2. Author is encouraged to provide the novelty of IL-36G to be used as a therapeutic target. I request to perform Receiver operating characteristic (ROC) analysis for IL-36G along with a reference marker.

Reviewer #2: The title:

- Meta-analysis may better describe an analysis of every single available transcriptomic dataset either microarray (which was done in this study) and RNA-seq data (which was not done in this study). Analysis of 2 datasets among the available ones may better be described by “analysis of transcriptomic data” instead of “meta-analysis.”

Line 57 (introduction):

- Liquefaction degeneration definition and characterization is missing specially it might be related to malignant transformation as mentioned in line 54 (PMID: 28556960).

Line 81 (methods):

- There are available public transcriptomic datasets such as GSE70665 (RNA-seq data). The present study is focusing on microarray data, so it might be better to skip mentioning the part that only 2 groups deposited their data as they their data as other data can be accessed in SRA format.

Line 98 (methods):

- “hclust function of the R package”: the package link is provided, but not the package name. It may be added “stats package in R”

Line 155 (figure 1 legend):

- It might be better to mention color change without gradual as the word gradual fits more for single cell transcriptomic data when a large number of cells is being plotted including cells during a transition state between the analysis conditions or when the data is a time-series one.

Line 156 (figure legend):

- It is very interesting that the authors marked the outliers in the heatmap (as you did) instead of plotting the final heatmap after removing outliers. Also, including how the downstream analysis resulted in no overlapping genes when all samples were included is very interesting.

Line 176 (results):

- Overlapping DEGs may need to be further described. Does it mean that the gene was found to be consistently upregulated or downregulated in both datasets OR the gene is considered to be overlapping if it was identified as a differentially expressed gene regardless the direction (upregulated or downregulated)? Was any genes found to be upregulated in a dataset and downregulated in the other dataset (bidirectional)? If yes, they should be mentioned. If no, that should be also mentioned.

Reviewer #3: Please re-check for minor grammatical inconsistency.

a) at line #134: please refine the header with a clear message of this result section

b) line 149: S2 Table-).

c) line #166: transcritomes should be transcriptomes

Reviewer #4: some minor concerns related to manuscript that needs to be addressed:

1. Why only IL-36G was chosen for validation at protein level while there were other potential candidate genes with higher fold change than IL-36G?

2. Authors should explain why the samples with other oral diseases such as chronic inflammation, acanthosis etc was chosen as controls in this study. Is it possible that the chronic inflammation in these control tissues are early signs of OLP?

3. IHC images needs to be labelled properly. It will help to emphasise the localisation of gene expression in specific cells or areas.

6. PLOS authors have the option to publish the peer review history of their article (what does this mean?). If published, this will include your full peer review and any attached files.

Reviewer #1: No

Reviewer #2: **Yes: **Ahmed S Abouhashem

Reviewer #3: No

Reviewer #4: **Yes: **Renu Bala

---

## [Author Response · Author response to Decision Letter 0]

16 Aug 2021

-> The style requirements have been reconfirmed.

“This study was supported by the National Research Foundation of Korea (Daejun, Korea) through the grants 2018R1A5A2024418 and 2020R1A2C2007038 awarded to Youngnim Choi and 2019R1A2C1002350 awarded to Sun Shim Choi.”

-> The funders had no role, and the suggested statement has been added.

“This study was supported by the National Research Foundation of Korea (Daejun, Korea) through the grants 2018R1A5A2024418 and 2020R1A2C2007038 awarded to Youngnim Choi and 2019R1A2C1002350 awarded to Sun Shim Choi.”

We note that you have provided funding information within the Acknowledgements. Please note that funding information should not appear in the Acknowledgments section or other areas of your manuscript. We will only publish funding information present in the Funding Statement section of the online submission form.

“This study was supported by the National Research Foundation of Korea (Daejun, Korea) through the grants 2018R1A5A2024418 and 2020R1A2C2007038 awarded to Youngnim Choi and 2019R1A2C1002350 awarded to Sun Shim Choi.”

-> The amended funding statement was removed from the manuscript and included in our cover letter.

-> The captions for Supporting Information files have been included at the end of manuscript.

Reviewers' comments:

Reviewer #1: While going through the manuscript, I came through some lacking which have been addresses below:

-> We thank the reviewer for constructive comments that improved the clarity of our manuscript.

1. I request the author to give logical and sound reasons for selection of IL-36G out of several overlapping DEGs identified. Out of two cytokines IL-36G and IL-1, author has chosen only IL-36G, I request to provide reason for selection of IL-36G.

-> We chose IL-36G among the top five overlapping DEGs. IL-1 belongs to DEGs in the epithelium dataset but not in the mucosa dataset. The reason has been added at lines 210-212 of the revised manuscript as follows: To validate the results of our bioinformatic analysis at the protein level, IL-36G was chosen among the top five upregulated overlapping DEGs because a secretory cytokine is an attractive therapeutic target compared with intracellular proteins.

2. Author is encouraged to provide the novelty of IL-36G to be used as a therapeutic target. I request to perform Receiver operating characteristic (ROC) analysis for IL-36G along with a reference marker.

-> The ROC analysis was performed as suggested. The result was added to Fig. 3 and Result section (lines 246-248) as follows: ROC curve analysis revealed that the expression levels of IL-36� both in the epithelium and lamina propria could differentiate OLP from disease controls based on the area under curve (AUC > 0.7, p < 0.05). In contrast, IL-36Ra was not a significant marker (Fig. 3f-h).

Reviewer #2: 

-> We thank the reviewer for constructive comments that improved the clarity of our manuscript.

The title:

- Meta-analysis may better describe an analysis of every single available transcriptomic dataset either microarray (which was done in this study) and RNA-seq data (which was not done in this study). Analysis of 2 datasets among the available ones may better be described by “analysis of transcriptomic data” instead of “meta-analysis.”

-> The title has been changed as suggested, and the expression “meta-analysis” throughout the manuscript has been removed.

Line 57 (introduction):

- Liquefaction degeneration definition and characterization is missing specially it might be related to malignant transformation as mentioned in line 54 (PMID: 28556960).

-> It has been added at lines 52-54 as follows: In particular, the liquefaction degeneration reflects senescence of attacked basal cells and resembles the typical epithelial-mesenchymal transition alteration, thus, it might be related to malignant transformation [4-6].

Line 81 (methods):

- There are available public transcriptomic datasets such as GSE70665 (RNA-seq data). The present study is focusing on microarray data, so it might be better to skip mentioning the part that only 2 groups deposited their data as they their data as other data can be accessed in SRA format.

-> We are sorry that we missed a precious dataset from our search. The sentence has been edited as follows: Among the five previous studies, two transcriptome datasets, GSE52130 [10] and GSE38616 [11], deposited in public databases were included in the present study.

Line 98 (methods):

- “hclust function of the R package”: the package link is provided, but not the package name. It may be added “stats package in R”

-> “the hclust function of the R package” has been changed into “the hclust stats package in R”.

Line 155 (figure 1 legend):

- It might be better to mention color change without gradual as the word gradual fits more for single cell transcriptomic data when a large number of cells is being plotted including cells during a transition state between the analysis conditions or when the data is a time-series one.

-> “gradual” has been removed.

Line 156 (figure legend):

- It is very interesting that the authors marked the outliers in the heatmap (as you did) instead of plotting the final heatmap after removing outliers. Also, including how the downstream analysis resulted in no overlapping genes when all samples were included is very interesting.

-> The final heatmap after removing outliers is same with the one before removing outliers (below). We wanted to save the space by skipping redundant data. The mucosa dataset seemed to vary a lot from sample to sample, probably due to variation in the degree or composition of immune cell infiltration. Because removing outliers strengthens the power of detecting DEGs, we could identify more overlapping DEGs. 

Line 176 (results):

- Overlapping DEGs may need to be further described. Does it mean that the gene was found to be consistently upregulated or downregulated in both datasets OR the gene is considered to be overlapping if it was identified as a differentially expressed gene regardless the direction (upregulated or downregulated)? Was any genes found to be upregulated in a dataset and downregulated in the other dataset (bidirectional)? If yes, they should be mentioned. If no, that should be also mentioned.

-> Overlapping DEGs were further specified at lines 173-175 as follows: In the comparison of the DEGs of the two partial datasets, 43 overlapping DEGs (23 upregulated and 20 downregulated in both sets) were identified (Fig. 1h and Table 1). There was no common DEG that was upregulated in one set but downregulated in the other set.

Reviewer #3: Please re-check for minor grammatical inconsistency.

-> We thank the reviewer for constructive comments that improved the clarity of our manuscript.

a) at line #134: please refine the header with a clear message of this result section

-> The header has been revised as follows: Removing outliers increased the number of DEGs in each dataset, and 43 overlapping DEGs were identified

b) line 149: S2 Table-).

-> Corrected.

c) line #166: transcritomes should be transcriptomes

-> Corrected.

Reviewer #4: some minor concerns related to manuscript that needs to be addressed:

-> We thank the reviewer for constructive comments that improved the clarity of our manuscript.

1. Why only IL-36G was chosen for validation at protein level while there were other potential candidate genes with higher fold change than IL-36G?

-> The reason has been added at lines 210-212 of the revised manuscript as follows: To validate the results of our bioinformatic analysis at the protein level, IL-36G was chosen among the top five upregulated overlapping DEGs because a secretory cytokine is an attractive therapeutic target compared with intracellular proteins.

2. Authors should explain why the samples with other oral diseases such as chronic inflammation, acanthosis etc was chosen as controls in this study. Is it possible that the chronic inflammation in these control tissues are early signs of OLP?

-> The main reason for using controls with other oral diseases was availability. Among the tissue blocks stored in the tissue bank, cases with minimal histopathological abnormality were selected. In the aspect of evaluating biomarkers, however, the use of disease controls is important. The chronic inflammation observed in 3 control tissues was scattered throughout the lamina propria rather than presenting a band-like pattern close to the epithelium. Therefore, it is not likely to be early signs of OLP. The selection of control tissues was explained at lines 217-218 and 227-228 as follows: The expression of IL-36g was examined by immunohistochemistry using tissue sections of 25 OLP cases and 7 control cases with other oral diseases that were chosen based on the histopathology and availability. The histopathological diagnoses of the control tissues included chronic inflammation, acanthosis, fibroma, epulis fissuratum, and hyperkeratosis, but the histopathological abnormalities observed in control tissues were limited.

3. IHC images needs to be labelled properly. It will help to emphasize the localization of gene expression in specific cells or areas.

-> It has been added at lines 234-235 and 239 as follows: As depicted in Fig. 3a, IL-36g was expressed in epithelial cells (asterisks) throughout the epithelium of OLP tissue and also in the infiltrated immune cells (arrows). The expression pattern of IL-36Ra was similar to that of IL-36g (Fig. 3c).

---

## [Decision Letter · Decision Letter 1]

31 Aug 2021

Gene signatures associated with barrier dysfunction and infection in oral lichen planus identified by analysis of transcriptomic data

PONE-D-21-21443R1

Dear Dr. Choi,

We’re pleased to inform you that your manuscript has been judged scientifically suitable for publication and will be formally accepted for publication once it meets all outstanding technical requirements.

Kind regards,

Kanhaiya Singh, Ph.D

Academic Editor

PLOS ONE

Additional Editor Comments (optional):

Reviewers' comments:

Reviewer's Responses to Questions

**Comments to the Author**

1. If the authors have adequately addressed your comments raised in a previous round of review and you feel that this manuscript is now acceptable for publication, you may indicate that here to bypass the “Comments to the Author” section, enter your conflict of interest statement in the “Confidential to Editor” section, and submit your "Accept" recommendation.

Reviewer #1: All comments have been addressed

Reviewer #2: All comments have been addressed

Reviewer #4: All comments have been addressed

2. Is the manuscript technically sound, and do the data support the conclusions?

Reviewer #1: Yes

Reviewer #2: Yes

Reviewer #4: Yes

3. Has the statistical analysis been performed appropriately and rigorously? 

Reviewer #1: Yes

Reviewer #2: Yes

Reviewer #4: Yes

4. Have the authors made all data underlying the findings in their manuscript fully available?

Reviewer #1: Yes

Reviewer #2: Yes

Reviewer #4: Yes

5. Is the manuscript presented in an intelligible fashion and written in standard English?

Reviewer #1: Yes

Reviewer #2: Yes

Reviewer #4: Yes

6. Review Comments to the Author

Reviewer #1: While reviewing the manuscript I found that all the suggestions have been beautifully addressed by the author and can be accepted for publications.

Reviewer #2: (No Response)

Reviewer #4: (No Response)

7. PLOS authors have the option to publish the peer review history of their article (what does this mean?). If published, this will include your full peer review and any attached files.

Reviewer #1: No

Reviewer #2: **Yes: **Ahmed S Abouhashem

Reviewer #4: No

---

## [Editor Report · Acceptance letter]

2 Sep 2021

PONE-D-21-21443R1 

Gene signatures associated with barrier dysfunction and infection in oral lichen planus identified by analysis of transcriptomic data 

Dear Dr. Choi:

I'm pleased to inform you that your manuscript has been deemed suitable for publication in PLOS ONE. Congratulations! Your manuscript is now with our production department. 

Kind regards, 

on behalf of

Dr. Kanhaiya Singh 

Academic Editor

PLOS ONE